# Acceptability of a community health worker-led health literacy intervention on lifestyle modification among hypertensive and diabetes patients in the City of Harare, Zimbabwe

**Nyaradzai Arster Katena**[1]*, **Shepherd Shamu**[1], **Golden Tafadzwa Fana**[2], **Evans Dewa**[3], **Admire Dombojena**[3], **Simbarashe Rusakaniko**[1]

1 Department of Global Public Health and Family Medicine, University of Zimbabwe, Harare, Zimbabwe, 2 Department of Medicine, University of Zimbabwe, Harare, Zimbabwe, 3 Department of Health Service, City of Harare, Harare, Zimbabwe

* nyarikatena@gmail.com

## Abstract

Working with community health workers is a vital strategy to improve health at a community level in low- and middle-income countries. Our study assessed the acceptability of a community health worker-led health literacy intervention on lifestyle modification among hypertensive and diabetes patients in the City of Harare, Zimbabwe. The intervention consisted of face-to-face individual educational sessions and support visits, delivered by trained community health workers at either the patient's home or the primary care clinic. We embedded this qualitative study within a cluster randomized trial, which assessed the intervention's effectiveness. Data were gathered through in-depth interviews with 3 community health nurses and 25 patients as well as 3 focus group discussions with CHWs. We analyzed the data manually using the deductive analysis method based on a coding framework structured according to the Theoretical Framework for Acceptability. Participants expressed optimism and anxiety regarding the intervention. All community nurses and CHWs believed that the intervention was effective in improving adherence to recommended lifestyle modifications and overall health outcomes among the patients. Patients felt that the intervention was ethically sound. All community health workers and community nurses reported a clear understanding of the intervention's goals and methods. Some patients felt that some aspects of the intervention needed to be improved. There was a consensus that the benefits associated with the intervention outweighed the costs. Some patients reported that they were not confident in participating in the intervention because some of the recommended lifestyle modifications were beyond their control. Whilst there were positive sentiments regarding the intervention's potential to empower patients and improve health outcomes, challenges related to patients' perceived burdens must be addressed. Future iterations of the intervention should focus on enhancing support for CHWs and ensuring that patients' preferences are catered for.

**Data availability statement:** All relevant data are within the paper and its Supporting information files.

**Funding:** Research reported in this publication was financially supported by the Fogarty International Center and National Institute of Dental and Craniofacial Research of the National Institutes of Health under Award Number D43 TW011968 to NAK. The content is solely the responsibility of the authors and does not necessarily represent the official views of the National Institutes of Health.

**Competing interests:** The authors have declared that no competing interests exist.

## Introduction

Working with community health workers (CHWs) is a vital strategy to improve health at a community level in LMICs [1–4]. Community health workers are voluntary public health workers who play a crucial role in primary and tertiary prevention of diseases [5]. Working with CHWs in the implementation of public health interventions can facilitate the linkage between communities and health services and thus ensure community participation. For many years, CHWs in LMIC, particularly in Sub-Saharan Africa, have been providing community-based care, but their efforts have been mainly focused on home-based caring for people living with HIV and TB, [5,6]. However, the contribution of CHWs to infectious chronic disease care in many low- and middle-income countries (LMICs) suggests that they could play a significant role in the management of non-communicable diseases(NCDs). Management of NCDs in these countries is often poor, with health systems struggling with several challenges that include staff shortages [7]. There has been an increase in the burden of NCDs in LMICs over the recent years. Worldwide, NCDs kill 41 million people each year, which is equivalent to 74% of all deaths globally [6,7]. Each year, 17 million people die from an NCD before the age of 70, and 86% of these premature deaths occur in LMICs. Overall, 77% of all NCD deaths are in LMICs. Amongst all NCDs, cardiovascular diseases account for most NCD deaths, with 17.9 million deaths annually [8,9].

Diabetes and hypertension are amongst the most prevalent NCDs worldwide and CHW-led interventions for these conditions are increasingly becoming popular in LMICs [10]. In the countries where the CHWs participated in the management of these NCDs, their roles included education and awareness, screening and monitoring, support for medication and lifestyle modification adherence, referral to healthcare professionals, and advocating for healthier environments [5,11,12]. Overall, CHWs provide additional workforce for the management and care of hypertension and diabetes patients where resources are limited and allow for task shifting to alleviate overstretched health systems [13,14]. Community health workers can play a beneficial role in improving health outcomes among diabetes and hypertensive patients while improving equity in health care delivery [15]. As a result, it is increasingly recommended that services be offered outside of health facilities in communities, to address the rising burden of hypertension and diabetes.

Several studies have reported positive outcomes for CHW interventions for diabetes and hypertension, such as reduced cardiovascular risk, improved medication adherence, improved lifestyle modification, and improved quality of life [3,16–20]. For instance, in a scoping review of 54 articles on the role of community health workers in type 2 diabetes mellitus self-management, it was concluded that CHWs play several roles, including structured education, ongoing support, and health system advocacy [21]. Another review of studies evaluating the effectiveness of community-based interventions for the prevention of Type 2 diabetes in LMICs also confirmed the successful work of CHWs in the management of type 2 diabetes [22]. In a participatory qualitative research project carried out by Chimberengwa and Naidoo, it was concluded that in the management of hypertension, CHWs were a key link between the community and the formal health delivery care system [20].

In Zimbabwe, CHWs provide community-based care, but like in other LMICs, for years their efforts have been focused on home-based care for people with HIV and TB, with little done around NCDs [23]. This shows the need to continuously come up with effective CHW-led interventions for the management of common NCDs. This paper reports on a qualitative study to assess the acceptability of a community health worker-led health literacy intervention on lifestyle modification among hypertensive and diabetes patients in the City of Harare, Zimbabwe. The study was embedded in a cluster randomized trial to assess the effectiveness of the intervention.

## Methods

### Study setting

The study was conducted in three City of Harare primary health care clinics, which were in the intervention arm of the cluster randomized controlled trial. The 3 health facilities are part of 39 health facilities (12 polyclinics, six family health services clinics, 15 satellite clinics, and two infectious diseases hospitals) in the city. The facilities offer comprehensive primary health care services, which include curative services, care of chronic patients, maternal and child health services, human immune-deficiency(HIV) prevention services, and community health services. These clinics are located in the high-density suburbs of the city, with each clinic catering to a population of 80,000–100,000 people, who are mostly socio-economically disadvantaged.

### Study design

Our study employed the qualitative design.

### Description of the intervention

Our intervention was a simple health literacy intervention on lifestyle modification for patients with diabetes and hypertension and it was delivered by trained community health workers. The intervention was based on the constructs of the Health Belief Model, and it aimed to: (i) convey the consequences of the patient's poor adherence to recommended lifestyle modifications; (ii) communicate to the patient the list of recommended lifestyle modifications and highlighting the benefits (iii) provide assistance in identifying and reducing barriers to adopt the recommended lifestyle modifications and (iv) develop cues to action for the patient.

The intervention consisted of face-to-face individual educational sessions and support visits. Four educational sessions were conducted with each patient at either the patient's home or the primary care clinic. The first sessions were held at the clinic on the day of recruitment into the study. The second and third sessions were held at either the clinic or at home as preferred by the patient. The fourth sessions were home visits, where the CHW visited the patient in their homes as a way of support and follow-up. After the fourth educational session, the CHWs continued to visit the patients at their homes monthly. In the first session, the CHW educated the patient on the recommended lifestyle modifications and the consequences of non-adherence. In the second session, the focus was on the patients' barriers to adopting recommended lifestyle modifications and ways to overcome them. Additionally, the patients wrote down their plans for lifestyle modification on a card that they stuck at a visible place in their houses. The cards were meant to serve as reminders to perform the planned actions. In that regard, the second sessions were highly interactive, when compared to the first sessions. The third and fourth sessions were supportive and evaluation sessions where the CHWs assessed whether the patients had been following their plans. Any deviations from the plan were discussed and ways to correct them were explored. In the fourth sessions, available family members were involved to support the patient, and this was also meant to serve as another cue to action for the patient. The monthly home visits that were conducted after the fourth educational session served as support visits where the CHWs continued to reinforce the importance of adhering to lifestyle modifications. The first sessions were conducted on the day of recruitment, the second sessions were held a week after the first sessions. The third and fourth sessions were conducted after two weeks thereafter. Each educational session (sessions 1–4) lasted for 30–45 minutes, whilst the monthly support visits lasted between 10–15 minutes. A protocol for the cluster randomized controlled trial that contains full details of the intervention implementation has been published in the JMIR research protocols journal [24].

## Description of the community health workers

The 15 CHWs who were involved in the study were selected from a group of over 200 CHWs who operate in the city of Harare. They have been serving as CHWs in the City for periods that range between 5–30 years. The CHWs in the city range from young adults in their mid-twenties to older individuals in their fifties or sixties. Ninety-eight percent of the CHWs are females and most of them are married. In terms of education levels, all the CHWs in the City can read and write and they have gone up to primary or secondary school level. They work 12 days per month and on each day, they work for 4 hours (from 8 am up to noon). Though they are volunteers, they receive a modest monthly stipend from the City of Harare Health Services Department. From time to time, they also receive monetary, and equipment support from various non-governmental organisations, to cater for their outreach work. The CHWs play a crucial role in the health care delivery in the city. They serve as a bridge between the healthcare system and the community, ensuring that everyone has access to the information they need to stay healthy. On recruitment, all the CHWs undergo an eight-week training offered by the Ministry of Health and Childcare. They also undergo refresher trainings when the need arises, as provided for by the guidelines for CHWs in Zimbabwe [22,25]. To implement our intervention, the 15 CHWs attended a one-day training.

## Conceptual framework

Acceptability was assessed from the perspectives of the CHWs, community nurses, and patients, using Sekhon et al's Theoretical Framework of Acceptability (TFA) [25]. Sekhon et al. defined acceptability as "a multi-faceted construct that reflects the extent to which people delivering or receiving a healthcare intervention consider it to be appropriate, based on anticipated or experienced cognitive and emotional responses to the intervention" [25] The TFA comprises of seven constructs that are considered as the core indicators of acceptability. These constructs and their definitions are presented in Table 1 below.

## Study participants

A total of 43 participants participated in the study. These participants comprised of  the 3 community health nurses who were the supervisors of CHWs from the three study cites, 25 patients (6–8 patients per clinic) and 15 CHWs who implemented the intervention. The 25 patients who participated were selected using the purposive sampling technique and the sample size was not predetermined but reached after saturation of themes.

Table 1. Theoretical Framework of Acceptability (TFA) Constructs.

| Construct | Definition |
| --- | --- |
| Affective attitude | How an individual feels about the intervention |
| Burden | The perceived amount of effort that is required to participate in the intervention |
| Ethicality | The extent to which the intervention is a good fit with an individual's value system |
| Intervention coherence | The extent to which the participant understands the intervention and how it works |
| Opportunity costs | The extent to which benefits, profits, or values must be given up engaging in the intervention |
| Perceived effectiveness | The extent to which the intervention is perceived as likely to achieve its purpose |
| Self-efficacy | The participant's confidence that they can perform the behavior(s) required to participate in the intervention |

Source: Sehkon et al, 2017 [25].

## Data collection

Data was collected during the implementation of the intervention during the period 01 August 2023–31 January 2024 (i.e., a month after the start of the implementation up to the end of the implementation) through focus group discussions (FGDs) and in-depth interviews(IDIs). In-depth interviews were held with community health nurses and patients and FGDs were held with the CHWs. All the FGDs and IDIs were audio-recorded, using the researchers' smartphones. The principal researcher and 2 trained research assistants were involved in the data collection. We used a guide (S1 File) that was in the vernacular language (ChiShona) to conduct interviews and discussions. The questions were based on the conceptual framework, described earlier in this section. Examples of questions that were included in the guides were (1) what are the main challenges you met in implementing the intervention? (2) what are the benefits of implementing this intervention? (3) how did this intervention help you? (4) what did you like about the intervention? and (5) what did you not like about the intervention? In-depth interviews with community health nurses were conducted in a private space at the local clinic, whilst the interviews with patients were conducted at their homes. Focus group discussions with CHWs were conducted at the clinics.

## Ethical considerations

Ethics approval was obtained in two stages. The initial approval was granted by the Joint Research Ethics Committee for the University of Zimbabwe Faculty of Medicine & Health Sciences and the Parirenyatwa Group of Hospitals- JREC (Ref: JREC 339/ 2022) in September 2022. The JREC is the local institutional review board for the University of Zimbabwe, where the first author is based. This approval covered the period from 29 September 2022 to 28 September 2023. The purpose of this approval was to enable the researcher to do the preliminary fieldwork. The final approval was granted by the Medical Research Council of Zimbabwe – MRCZ (Ref: MRCZ/A/3059) in September 2023. The MRCZ is the national ethics committee in Zimbabwe. The approval from the MRCZ covered the period 04 September 2023 to 03 September 2024. This was the overall approval for all the final data collection. We also obtained permission to conduct the study in the City of Harare clinics from the Director of Health Services. Written informed consent was obtained from participants before data collection, and we maintained confidentiality at all stages.

## Data analysis

The deductive analysis method described by Braun & Clarke was used to analyze the data [26]. The deductive analysis is a theory-driven approach that deliberately explores data within the bounds of one or more theoretical frameworks [27]. Initial deductive coding was based on the seven constructs of the TFA, and inductive coding was used to explore new emerging themes that the TFA did not cover. The three researchers who were involved in the analysis listened to all recordings and deductively coded all transcripts separately using the coding sheet derived from the conceptual framework. The principal investigator then compared the sets of coded transcripts noting differences in coding. The differences were subsequently discussed and resolved by consensus.

## Results

### Characteristics of study participants

Table 2 below shows the characteristics of participants for the IDIs and FGDs.

**Table 2.** *Characteristics of study participants.*

| Category of participants | Sex | | Marital status | Age Description |
|---|---|---|---|---|
| | **Male** | **Female** | | |
| Patients | 11 | 14 | 13 married 6 widowed 1 single 5 divorced | Median (IQR):53 (42–68) years |
| CHWs | 0 | 15 | 11 married 3 divorced 1 widowed | Median (IQR): 42 (36–54) years |
| Community Nurses | 0 | 3 | All married | Median (IQR) 51 (49.5–65) years |

## Intervention acceptability

From the deductive analysis, the eight themes from the TFA were identified, with the subthemes. The themes and subthemes are presented below and supporting verbatim are shown in Table 3 below.

**Affective attitude.** The affective attitude construct revealed a spectrum of emotional responses of patients, community CHWs, and nurses towards the health literacy intervention. Patients expressed a mix of optimism and anxiety regarding the intervention. Some patients felt hopeful about getting knowledge and support from CHWs to modify their lifestyles and thus manage their conditions better. Negative emotional responses from patients included feelings of anxiety about the disruptions that the recommended lifestyle modifications, particularly exercising could bring to their daily routine. Other patients also reported feelings of worry about the delivery of the intervention. They believed that whilst the CHWs were trained to implement this intervention, they may fail to answer some of their questions as effectively as doctors and nurses. CHWs also expressed both negative and positive feelings towards the interventions. Some CHWs reported feelings of contentment, highlighting that the intervention helped them to fulfil one of their roles in the community, which is facilitating positive changes in patients' attitudes and health behaviors. Those CHWs who expressed negative emotions towards the intervention voiced concerns about the challenges of addressing diverse patient needs and the emotional toll of navigating resistance or skepticism from patients. Community nurses generally expressed gratefulness for the intervention highlighting its potential to enhance community engagement in the management of hypertension and diabetes.

**Burden.** The burden construct reflected the perceived challenges and demands associated with the intervention. The potential challenges of this intervention were mainly mentioned by the CHWs, whilst community nurses believed that it was quite feasible to implement the intervention. Some of the patients were a bit sceptical. The main challenges that were mentioned by CHWs included resistance by patients, insufficient resources, and lack of acceptance at some of the patients' homes. Overall, the CHWs reported experiencing a dual burden. On one hand, they felt the weight of their responsibility to effectively educate and motivate patients and on the other hand, they faced logistical burdens. There were some patients who expressed concerns about the perceived burden of additional responsibilities related to lifestyle modifications. They felt overwhelmed by the prospect of making significant changes to their daily routines, particularly regarding diet and physical activity. The community nurses expressed that there was no additional burden in implementing this intervention. They highlighted that the intervention lessened their burden of management of diabetes and hypertensive patients and promoted patient self-management.

**Table 3.** *Emerging Themes, Sub-Themes and Supporting Verbatim.*

| Theme (TFA Construct) | Subtheme | Group of Participants | Supporting Verbatim |
|---|---|---|---|
| Affective Attitude | Happy about the intervention's aim, content, and benefits | Patients, CHWs, Community Nurses | *"I feel hopeful about getting support to manage my condition."* Female Patient, IDI4<br>*" This intervention allows me to help my community."* CHW1, FGD2<br>*"I'm grateful for this intervention; it enhances our community engagement."* Community Nurse3<br>*"What's good about your intervention is that the CHWs live with us, so we can get information from them anytime, even on Sundays when you don't open your clinics".* Male Patient IDI23<br>*"This is indeed good. With the prevalence of diabetes and hypertension increasing, they are better managed at the community level, whilst we deal with cholera and other infectious diseases that are also highly prevalent in the City of Harare".* Community Nurse 2 |
|  | Anxious about the changes that result from participating in the intervention | Patients, CHWs | *"I'm anxious about how these changes will disrupt my routine."* Female Patient, IDI2<br>*"It is challenging to meet the diverse needs of my patients."* CHW2, FGD3 |
| **Burden** | Concerns about the need for additional resources to effectively implement the intervention and resistance of patients | CHWs | *"Patient resistance is a major hurdle for us."* CHW5,FGD2<br>*"We need more resources to effectively implement this."* CHW3, FGD 1<br>*While we live in the community, carrying out home visits to counsel and monitor patients on lifestyle modification is additional work which needs more resources like a bicycle and lunch allowance".* CHW1, FGD1 |
|  | Concerns about the effort needed to adopt recommended lifestyle modifications | Patients | *"I worry that these lifestyle changes will be too much for me."* Female Patient, IDI 15 |
|  | Beliefs that there are no challenges in implementing the intervention | Community Nurses | *" Implementing this intervention is not difficult There's no additional burden on us; actually, it reduces our workload."* Community Nurse 1 |
|  | Gratitude for free cost and easy access to information | Patients, CHWs | *"I'm grateful for free access to information."* Male Patient, IDI 24<br>*"For diabetes and hypertensive patients, going to the clinic to get information only is a mammoth task because they need to pay a consultation fee, USD5, therefore this intervention will also benefit those without money"* CHW5, FGD3 |
|  | Beliefs that the intervention is efficient | Community Nurses, Patients | *"Coordinating with CHWs to monitor and support diabetes and hypertensive patients is quicker than counseling patients myself."* Community Nurse 2<br>*"This intervention is very good. It will make us save money. We will no longer need to go to the clinics to ask about how to manage our conditions."* Female Patient, IDI 19 |
| **Perceived Effectiveness** | Beliefs that the intervention will empower patients through improved knowledge, skills, and practices to manage diabetes and hypertension effectively | Patients, CHWs, and Nurses | *"Lifestyle modification is an important component of managing hypertension and diabetes, but some of our patients do not know that. Therefore, this intervention will help people to understand."* Community Nurse2<br>*"Teaching people is indeed helpful. We have done that in other programs such as the Friendship Bench. Those programs were quite useful. Therefore, this intervention will be also effective".* CHW4, FGD3<br>*"Teaching the patients empowers the patients leading to better self-management and fewer complications."* Community Nurse 2<br>*"The guidance from CHWs has helped me understand my health."* Female Patient, IDI 7<br>*This intervention is good because when the CHW came to my home to teach me, my wife, and my grandchildren whom I stay with were also involved. They now know what I should do. This helps a lot because they now remind me to take the recommended actions"* Male Patient, IDI 10 |
|  | Concerns that the intervention is not sufficient to lead to behavior change | Patients | *"I believe medication is more important than lifestyle changes."* Male Patient, IDI 13<br>*"Interventions that focus on teaching people do not work. Only medication helps sick people".* Male Patient IDI 22 |
| **Ethicality** | Beliefs that the intervention was in tandem with social norms, religion, culture, and values | Patients, CHW, Nurses | *"You did well by making our CHWs teach us about the food that we should eat when we are diagnosed with diabetes or hypertension. They give us examples of the food that is available in our communities, unlike your doctors who will say eat food that is not available in our society"* Female Patient, IDI 16<br>*"When I teach the patient, I would have known already if his/ her religion does not allow some foods, so I will not mention those as examples of foods that should be eaten even if they are the recommended ones. I will talk about other acceptable substitutes".* CHW3, FGD2<br>*"These ladies (CHWs) will come up with better educational messages than us because they know what the community wants, values, and needs".* Community Nurse1 |
|  | Concerns about lack of privacy during home visits | Patients, CHWs | *"Some patients preferred private meetings away from home."* CHW, FGD 3<br>*"As for me, I am not comfortable with the CHWs coming to my home, they will notice my poverty. I would rather go to their home to receive counseling because they are public figures, so their homes are open to everyone".* Female Patient IDI 14 |
|  | Fear of being coerced to change lifestyles. | Patients | *"I fear that I will end up complying with the advice of these ladies (CHWs) because they now continuously monitor us".* Female Patient, IDI 7 |

*(Continued)*

**Table 3.**  (Continued)

| Theme (TFA Construct) | Subtheme | Group of Participants | Supporting Verbatim |
|---|---|---|---|
| **Intervention Coherence** | Beliefs that the intervention goal, processes, and content were clear | Patients, CHWs, and Community Nurses | "*The training clarified our objectives for implementing this intervention. It's like having a roadmap that guides our actions*" CHW5, FGD 1<br>"*I appreciate how straightforward the program is. The information is very clear, and I know exactly what to expect, which makes it easier for me to participate.*" Female Patient IDI 19<br>*The clarity of the intervention's goals and processes allows us to provide better support to both community health workers and patients. It creates a unified approach to care.*" Community Nurse 3. |
|  | Need for more personalized educational sessions. | Patients | "*I drink beer and smoke cigarettes, so I would appreciate more tailored advice for my situation so that I can be able to quit.*" Male Patient, IDI 22 |
|  | Beliefs that the structure of the educational sessions and the content of each session were aligned well | Community Nurses, Patients, and CHWs | "*The four sessions are connected well. Each session has its objective. So it's easy to deliver*" CHW 4, FGD 2<br>"*I found that the sessions flowed well. The content was not just random; it all tied together, making it easier to understand how everything fits into my care.*" Female Patient, IDI8<br>"*The educational sessions created a more engaging learning environment and left everyone with a clear understanding*" Community Nurse 1 |
|  | Beliefs that the aim, content, and delivery of the intervention were in tandem with the other interventions for diabetes and hypertension | Community Nurses, Patients | "*Earlier this year we attended a workshop on the PEN strategy, I feel that your intervention complements the existing programs for diabetes and hypertension. The goals are aligned, so we're all working toward the same outcomes.*" Community Nurse 1<br>"*The information we received feels the same as what we also read on the WHO and Ministry of Health internet. It's reassuring to know that you are doing things that are the same as those important health organizations*". Female Patient. IDI 25 |
| **Opportunity Costs** | Beliefs that participation in the intervention will consume time for other activities | Patients and CHWs | "*This intervention takes time away from other important community initiatives.*" CHW3, FGD2<br>"*I fear that attending sessions will take time away from my family*". Female Patient, IDI 1 |
| **Self-efficacy** | Feelings of confidence to implement the intervention adopt recommended lifestyle modifications | Patients, CHWs, and Community Nurses | " *After learning from the ladies (CHWs), I feel more capable of making these lifestyle changes now.*" Male Patient, IDI13<br>"*….of cause we will be able to teach the patients. With this training we have received its possible.*" CHW 2, FGD2<br>"*We feel confident in supporting CHWs during this intervention.*" Community Nurse 2 |
|  | Feeling of fears and doubt that some of the recommended lifestyle modifications, particularly dietary related are beyond patients' control | Patients | "*As much as I want to eat the right food that the CHWs teach us, sometimes fear I can't find it, especially fruits. I don't have an orchard; my yard is small*". Male Patient, IDI20<br>"*Some changes feel out of my control.*" Female Patient, IDI 1 |

**Perceived effectiveness.**  All community nurses and the CHWs believed the intervention was effective in improving adherence to recommended lifestyle modifications and overall health outcomes among the patients. However, when it came to the patients, some believed the intervention was not effective. Patients who largely expressed positive perceptions regarding the effectiveness of the intervention felt that the guidance from CHWs was instrumental in helping them to understand their health conditions and the necessary lifestyle changes. The patients reported that personalized education and support increased their confidence in managing their diabetes and hypertension. Some male patients also believed that the intervention would successfully empower them to get support from their families in managing hypertension and diabetes because of the involvement of other family members in the counseling sessions. Those patients who believed that the intervention was not effective highlighted that when it comes to managing chronic conditions, behavior change is not as important as medication. Community health workers viewed the intervention as effective, particularly in fostering a trusting relationship with patients. They believed that their role in providing tailored education and support visits significantly contributed to patients' health improvements. They highlighted success stories in previous related interventions, which reinforced their belief in the intervention's impact. Community nurses also recognized the

potential effectiveness of the intervention, noting that empowering patients with health literacy could lead to better self-management and fewer complications. They appreciated the involvement of family support in the intervention and how it aligned with their goals of holistic patient care.

**Ethicality.**  All categories of participants believed that since the intervention is implemented by CHWs, it helped in addressing cultural issues important to lifestyle modification. Patients generally felt that the intervention was ethically sound, appreciating the emphasis on respect for their community values and norms, particularly on education regarding diet related lifestyle modifications. The patients expressed that the CHWs acted in their best interest by teaching them about the locally available foods they are expected to eat to manage hypertension and diabetes effectively. However, some patients raised concerns about the potential for coercion. There are CHWs who highlighted that there are some patients who were reluctant to be visited at their homes but prefer to meet the CHWs at other places such as the CHW's home, the clinic, or even at church because they want to maintain their privacy. The CHWs viewed their roles as ethically vital, as they aimed to promote health equity within the community. They also highlighted the importance of balancing teaching patients about lifestyle modification with respect for patient choices.

**Intervention coherence.**  The intervention coherence construct highlighted the participants' understanding and perception of the intervention's relevance, purpose, and clarity. All community health workers and community nurses reported a clear understanding of the intervention's goals and methods. Most of the patients indicated that the education and instructions that were provided by the CHWs were straightforward to follow. The patients noted that the structured approach of the intervention helped them to understand and adopt the recommended lifestyle modifications. However, some patients expressed a desire for more personalized content that directly addressed their unique circumstances. CHWs reported a strong sense of coherence regarding the intervention, emphasizing that the training they received helped them understand. They also believed that the notes they had allowed them to communicate confidently with patients. Community nurses felt that the intervention's objectives aligned well with the overall goals of patient-centered care and chronic disease management. They also appreciated the training of the CHWs before the implementation of the intervention.

**Opportunity costs.**  Among CHWs and patients, there was a concern that participation in the intervention will consume time for other activities. The CHWs highlighted that the intervention could limit the time they had available for other community health initiatives, whilst patients highlighted that they will need to take some time from their household chores to participate in the intervention.

**Self-efficacy.**  Most of the patients reported feeling more confident in their ability to make sustainable lifestyle modifications after receiving education from CHWs. Some patients reported that they were not confident in participating in the intervention because some of the recommended lifestyle modifications (such as the consumption of fruits and vegetables every day) were beyond their control. CHWs felt confident in their roles, believing that their training and skills as well as the training they received before implementation of the intervention enabled them to effectively support patients. Nurses reported confidence in offering support to CHWs as they implemented the intervention.

## Discussion

As the burden of NCDs continues to rise worldwide, LMICs continue to struggle to manage these conditions due to several challenges, including staff shortages [28]. One emerging global strategy to address staff shortages in the management of chronic conditions is partnering with

CHWs to reach vulnerable populations in under-resourced settings [29]. Several reviews have found that over 60% of CHWs in LMIC performed some sort of home-based care; however, most of these efforts focused on HIV/AIDS, immunization, and cancer, and their work has also been mainly in rural areas [30,31]. Expanding on this work, we conducted a cluster randomised control trial to assess the effects of community health worker-led health literacy intervention for lifestyle modification among hypertensive and diabetes in an urban area in Zimbabwe. To enhance our assessment of the intervention, we embedded a qualitative acceptability study on this cluster randomized trial. Our study aimed to explore the acceptability of the intervention, using the seven acceptability indicators described by Sekhon et al. [25] in their TFA.

Acceptability has increasingly become a key consideration in the design and implementation of public health interventions [25]. It has been highlighted as one of the necessary conditions for the effectiveness of an intervention [25,32]. For instance, if an intervention is considered acceptable by patients, they are more likely to adhere to the recommendations and thus benefit from improved clinical outcomes. Likewise, if healthcare providers consider an intervention as acceptable, they are more likely to deliver it as intended, which also enhances the effectiveness of the intervention. Findings from our study revealed valuable insights from patients, community health workers, and nurses. The findings encompass a spectrum of both negative and positive perceptions and experiences. These findings provide a comprehensive understanding of stakeholders' attitudes, which are essential for refining and enhancing the intervention.

The emotional responses of all categories of participants underscores the complex nature of public health interventions. Patients expressed a duality of optimism and anxiety, revealing that while they were hopeful about gaining knowledge and support from CHWs, they also grappled with concerns about how lifestyle changes could disrupt their daily lives. This finding aligns with existing literature that emphasizes the emotional burden often experienced by patients managing chronic conditions [32,33]. For CHWs, the sense of fulfillment derived from facilitating positive changes indicates a strong intrinsic motivation that could enhance the intervention's efficacy. However, the emotional toll of navigating patient resistance points to the need for ongoing support and training for CHWs to manage these dynamics effectively. The importance of training has been highlighted as one of the key facilitators of the acceptability of CHWs in a rural area in South Africa [2].

The burden construct revealed that while CHWs and nurses viewed the intervention as feasible, patients expressed skepticism about the additional responsibilities associated with lifestyle modifications. This discrepancy highlights a critical aspect of intervention design, which is the need to balance the expectations placed on patients with the practical realities they face in their daily lives. The dual burden reported by CHW (feeling responsible for educating patients while facing logistical challenges) suggests that support structures should be implemented to alleviate their workload and enhance their capacity to engage effectively with patients. The need to put support structures for CHW-led interventions has also been reported elsewhere [2].

All participants recognized the intervention's potential to improve health outcomes. Patients who felt positively about the intervention attributed their increased understanding and confidence in managing their conditions to the personalized support from CHWs. Those who were skeptical of the intervention's effectiveness often prioritized medication over lifestyle changes, reflecting a common perception in chronic disease management. This finding emphasizes the need for interventions to not only focus on lifestyle changes but also to integrate medication adherence and other forms of social supports into the educational components, thus presenting a more holistic approach to managing diabetes and hypertension, as highlighted in previous studies [3,34,35].

For CHW – led interventions, it has been noted that consideration of how information is presented and the importance of fostering an environment where patients feel supported in making informed choices is critical, to ensure that patients feel empowered rather than coerced into making healthy choices [32,34]. Findings from our study were aligned with these previous findings. Patients raised fears about being coerced to make changes. Apart from concerns about being coerced, the ethical considerations surrounding our intervention were notable, particularly regarding cultural sensitivity and respect for patient choices. The alignment of CHWs' cultural backgrounds with those of the patients was perceived as an advantage in addressing community-specific health needs. The coherence of the intervention was largely affirmed by CHWs and nurses, suggesting a shared understanding of its objectives and methods. Patients' desire for more personalized content indicates a need for adaptability within the intervention to cater to individual circumstances. This finding which suggests that tailoring interventions to meet diverse patient needs can enhance engagement and adherence, ultimately leading to better health outcomes also aligns with the existing body of literature [31,36–38].

The recognition that the benefits of the intervention outweighed the costs indicates a positive perception of its value within the community. Patients appreciated the accessibility of health information, and the lack of financial burden associated with the intervention. However, the potential impact on their time and daily responsibilities underscores the need for interventions to be mindful of patients' broader life contexts. The improvement in self-efficacy reported by patients is a crucial indicator of the intervention's potential success. By enhancing confidence in their ability to implement lifestyle changes, the intervention may foster greater long-term adherence to health recommendations. However, addressing the concerns of patients who feel limited by external constraints is essential, particularly when the adoption of dietary related lifestyle modification is concerned. Ongoing support and resources should be made available to empower all patients, particularly those facing significant barriers to change. Interventions that offer support to patients (such as provision of nutritious food or skills to locally produce nutritious food) have been reported to be effective elsewhere [34,35,39,40].

Overall, findings from our study indicated that whilst our intervention was acceptable, there is room for strengthening on scaling up. Our study had notable strengths. We conducted the study during the implementation of the intervention. Therefore, recall bias was minimized since the participants were still recalling their experience with the implementation of the intervention. We assessed the acceptability from the perspectives of all those categories of people who are key to the success of the intervention (the beneficiaries- patients, the implementers- CHWs, and the supervisors- community nurses). Therefore, our findings are quite rich. Nevertheless, there are some limitations to our study. We cannot rule out the possibility of social desirability bias, particularly from the patients and the CHWs. There is a possibility that they could not have highlighted the potential challenges of the intervention because the principal investigator was involved in the conduct of the in-depth interviews and the focus group discussions. We tried to minimize this by limiting the role of the principal investigator to recording responses only. Two research assistants who were not part of the intervention were the ones who interviewed participants and facilitated focus group discussions. Secondly, results on the perceived effectiveness could be limited because the data was collected during the implementation of the intervention, such that the ultimate benefit may not have been experienced. To this end, we recommend a post-acceptability assessment. Nonetheless, the results of this study provide valuable information for upscaling of the intervention. Based on the patient's perceptions, we also recommend that on upscaling our intervention, considerations should be made to explore the provision of nutritious food to patients or empower them to produce nutritious food locally.

## Conclusion

The findings from this qualitative study highlight the multifaceted perceptions of acceptability surrounding the CHW-led health literacy intervention for lifestyle modification among hypertensive and diabetes patients. While there are positive sentiments regarding its potential to empower patients and improve health outcomes, challenges related to emotional responses, perceived burdens, and the need for personalized content must be addressed. Future iterations of the intervention should focus on enhancing support for CHWs, integrating medication management into lifestyle education, and ensuring that patient choices remain at the forefront of the intervention's ethical framework. By doing so, the intervention can better align with the needs and realities of the community it aims to serve. Future research should focus on assessment of the long-term impact of the intervention and improving access to recommended food for the management of hypertension and diabetes.

## Supporting information

**S1 File.  In-depth Interviews and Focus Group Discussion Guide.**
(DOCX)

**S2 File.  Acceptability transcripts.**
(DOCX)

## Acknowledgments

We would like to acknowledge the City of Harare, City Health Director for granting permission to carry out the study in the city of Harare health facilities. We also extend our gratitude to the study participants (diabetes and hypertension patients, CHWs, and community nurses) for their participation. We would also like to thank our colleagues from the Department of Global Public Health and Family Medicine for peer review of our study proposal.

## Author contributions

**Conceptualization:** Nyaradzai Arster Katena.

**Data curation:** Nyaradzai Arster Katena, Evans Dewa, Admire Dombojena.

**Formal analysis:** Nyaradzai Arster Katena, Evans Dewa, Admire Dombojena.

**Methodology:** Nyaradzai Arster Katena.

**Supervision:** Shepherd Shamu, Golden Tafadzwa Fana, Simbarashe Rusakaniko.

**Writing – original draft:** Nyaradzai Arster Katena, Shepherd Shamu, Golden Tafadzwa Fana, Evans Dewa, Admire Dombojena, Simbarashe Rusakaniko.

**Writing – review & editing:** Nyaradzai Arster Katena, Shepherd Shamu, Golden Tafadzwa Fana, Admire Dombojena, Simbarashe Rusakaniko.

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
