## [Decision Letter · Decision Letter 0]

3 Sep 2024

PGPH-D-24-01557

Acceptability of a community health worker- led health literacy intervention on lifestyle modification among hypertensive and diabetes patients in the City of Harare, Zimbabwe

Dear Author Team,

Thank you for submitting your manuscript to PLOS Global Public Health. After careful consideration, we feel that it has merit but does not fully meet PLOS Global Public Health’s publication criteria as it currently stands. Therefore, we invite you to submit a revised version of the manuscript that addresses the points raised during the review process.

The reviewers all recognized merit in the piece but Reviewers 2 and 3 have concerns about its current methodology and content that require significant revisions. Further consideration of the manuscript for submission is dependent on adequately addressing these, as I largely agree with the comments. Please carefully review Reviewer 2's methodological concerns, as well as the recommendations from Reviewer 3. As described below, please create a separate document where you respond to to each comment and detail how you did (or why you didn't) edit the manuscript as a result.

We look forward to receiving your revised manuscript.

Kind regards,

Bram Wispelwey, MD, MS, MPH

Academic Editor

Journal Requirements:

Additional Editor Comments (if provided):

Reviewers' comments:

Reviewer's Responses to Questions

**Comments to the Author**

1. Does this manuscript meet PLOS Global Public Health’s publication criteria ? Is the manuscript technically sound, and do the data support the conclusions? The manuscript must describe methodologically and ethically rigorous research with conclusions that are appropriately drawn based on the data presented.

Reviewer #1: Yes

Reviewer #2: Partly

Reviewer #3: Partly

2. Has the statistical analysis been performed appropriately and rigorously?

Reviewer #1: N/A

Reviewer #2: N/A

Reviewer #3: N/A

3. Have the authors made all data underlying the findings in their manuscript fully available (please refer to the Data Availability Statement at the start of the manuscript PDF file)?

Reviewer #1: Yes

Reviewer #2: Yes

Reviewer #3: No

4. Is the manuscript presented in an intelligible fashion and written in standard English?

Reviewer #1: Yes

Reviewer #2: Yes

Reviewer #3: Yes

5. Review Comments to the Author

Reviewer #1: This was a qualitative study looking at the acceptibility of an interventions . What makes this research unique is the focus on NCDs since most of the CHW interventions in sub-saharan africa are on HIV. The methodology was clearly defined that is qualitative research with focus groups and interviews. The authors explained well the frameworks used in research questions and the key themes in the analysis and conclusions. The table explaining the themes helped a lot. They also detailed the different ethical clearance respective to the context where the research was done and information about participant consent. The flow of the article was easy to follow and mentioned availability of data when needed. The limitations were well explained and information on the role of the investigators and power dynamics and possibility of social desirability bias

It would have been interesting to learn more about the selection criteria for the CHW and more information on their responsibilities, training they recieve and additional support they get then assignment to respective patients and if the patients selected were the same patients the CHW were responsible for

In terms of findings it is known based on literature to date that most CHW programs are acceptable because they are part of the community and understand the context well. It was important to note appreciation by patients of lifestyle information by CHW being more more useful and relatable compared to advice from patients especially knowledge on accessible food , understanding of religious and socio-economic context of the patients. This is important considering the importance of appropiate lifestyle adjustments in NCDs especially diabetes and hypertension and essential in designing CHW training and interventions

Reviewer #2: This manuscript describes a qualitative study nested within a cluster-randomized trial of a CHW intervention to promote positive lifestyle modifications for people with hypertension or diabetes. The stated goal of the qualitative study is to assess acceptability of the intervention during implementation. I think that qualitative research to understand implementation outcomes of interventions is important, and so I comment the authors for embedding this study within the larger trial. However, there are major methodological issues with this qualitative study that threaten its validity and make it difficult to publish in its current state. Unless I am misunderstanding many things, I think that it may be necessary to reconsider the entire conceptual framework, which basically means re-analyzing the data (for a deductive analysis). Specific points are raised below, divided into three sections: the most important comments related to the conceptual framework, which threaten validity; then some other major comments that could be addressed by revisions; and finally some more minor technical comments.

Major comments related to conceptual framework:

1) The authors state that they used three conceptual frameworks for acceptability to guide the deductive analysis. I appreciate the attempt to use a validated conceptual framework, but the conceptual framework used in this analysis is quite unclear. Combining multiple frameworks explaining the same concept (i.e. acceptability) is scientifically dubious – a framework is supposed to represent at consistent conceptual whole, so mixing and matching would threaten this validity. Also, it is not clear how the cited frameworks contributed. I am not seeing any sign of the Sekhon framework in this paper’s framework, which is the only one with a citation. The constructs are quite different and do not map. I cannot find the other two frameworks in the literature, nor do they have citations provided.

2) Taking the framework as stated, I am confused by the conceptual distinctions among the five constructs, and this confusion persists when I try to understand the results. (a)“Perceived feasibility” is defined as “beliefs about the practicality and ease of implementing the intervention,” but then “Perceived barriers” talks about the “obstacles, challenges,” which seems like it is just the opposite of practicality/ease of implementation. Conceptually, these are the same issue- just stated in a positive or negative form. On the other hand, “perceived barriers” also includes “negative consequences,” which is not a characteristic of the implementation process, but rather an effect of the intervention, so I am not sure why it is part of the same construct. (b) Another point of confusion is the relationship between perceived effectiveness and perceived benefits. If perceived benefits is about “positive outcomes of the intervention,” this seems highly related to perceived effectiveness, which is the intervention’s ability to achieve these positive outcomes.

3) Perhaps because of lack of clarity in the constructs of the conceptual framework, there is some mixing of the constructs in the results, which is confusing. For example, the first sentence of the “effectiveness” section says “One of the benefits that was mostly mentioned across all participant groups was effectiveness of the intervention.” (Line 256), which seems to be mixing the themes of benefits and effectiveness. In the “Barriers” section, the quote saying that people will attend the counseling but not practice what is taught is really about effectiveness, not an implementation barrier.

Other major comments:

1) The abstract says: “Whilst there is substantial evidence on the effectiveness of CHWs interventions, there is a need for more research on the mechanisms through which these interventions work. Understanding the acceptability of these interventions is one way of assessing the mechanisms through which they work.” I disagree that understanding acceptability leads to understanding of mechanism. Whether people find something acceptable or not does not tell you how it works. It may help to explain effectiveness results (e.g. low acceptability -> low uptake -> low effectiveness). It does not tell you how a behavioral intervention actually exerted an effect, which assumes that it was both accepted and effective.

2) The fifth paragraph of the introduction is poorly supported by the scientific literature. Please provide a citation for, “Whilst there is substantial evidence that CHW programs improve a range of health outcomes, these benefits tend to reduce or disappear when CHW programs are scaled up.” Also, please give evidence for the next sentence, which claims that low uptake is actually an issue for CHW programs. Alternatively, the authors may choose to eliminate this paragraph – I do not think it is necessary to justify studying acceptability.

3) “Our study revealed the 5 broad themes of acceptability under the conceptual framework. There were no additional themes that were derived from the analysis.” (Line 208) I am not sure this summary of the results is really appropriate for a deductive analysis. In a purely deductive approach, which is what is described, the authors started with a framework with 5 overarching themes, so by definition this is what they would end up with. A more relevant summary would be to briefly summarize the subthemes so that it is clear what are the perceived benefits, barriers, etc.

4) “However, amongst patients, there were mixed perceptions regarding the benefits of the intervention.” (Line 214). But all the quotes are about benefits. What were the mixed perceptions- why did people think it might not be beneficial? Representing negative opinions are important for qualitative research – otherwise the authors are just selecting the results that make the intervention look good.

5) Line 331 and final paragraph: How is the conclusion drawn that the intervention is generally acceptable, given that there were several important barriers to acceptability identified?

Technical comments

1) Quote attributions must be added to text. That is, each quote should be followed by a description indicating the coded identity of the speaker (e.g. Patient 1, patient 2) so that it is clear that the quotes are representing different participants and not any dominant participant. Also, attributions should give consistent information about participants – right now the attributions are inconsistent and in narrative, so some describe sex while others do not, some describe age while others do not, etc.

2) Do not repeat quotes in text and table – if the quotes are all going to be given in the table, there is no reason to narrate them in the text. If there are other quotes that support these themes, then the table is a good place to put them to demonstrate the richness of the data.

3) Table 3- the subthemes lack clarity because it is not clear who it applies to. For instance, “empowerment” actually refers to CHW empowerment, not patient empowerment. It is also not clear given the structure of the table which quotes support which subtheme. I suggest reformatting the table so that there are specific rows for each subtheme and its supporting quote, and there is more elaboration around what the subtheme actually is.

Reviewer #3: This is a report of a qualitative analysis embedded within a larger CHW program aiming to improve NCD outcomes in Zimbabwe. The paper has some benefits but will need significant edits to be eligible for inclusion in the peer-reviewed literature.

First the pros:

- This is an all-Africa team of authors who are based at an African university. They were able to plan a series of ambitious studies and get formal IRB review in-country for complex interventional trials with a control group.

- They planned this embedded qualitative study to gather greater insights on the acceptability of their intervention. They were able to get data from both providers and patients.

- The used a variety of existing and validated tools to study such acceptability.

These are the main cons I noted:

- As it currently exists, this paper adds little to the CHW literature on NCD-focused interventions and contains messages that may actually be potentially harmful. It is, however, salvageable so my comments are offered in the spirit of helping the authors present their efforts and findings with a different framing.

- There are some typos, so I recommend they have a proof-reader go over the paper carefully. For example, even the abstract is missing the work “of” in “assess the effectiveness [of] community…”

- I recommend they avoid the use of words like “deploy” CHWs or “use” CHWs. This cadre of health workers are people, not tools. Use words like “partner with” or “employ” to suggest they have agency and autonomy.

- Line 54 they claim that little is being done in NCDs. This simply isn’t true, and there is a huge literature around the effectiveness of CHW interventions for NCDs, with often very good results. I suggest they conduct a more careful literature review, and include Latin American and North America in their search. It is possible there is less being done in Africa currently, but there is still plenty to review. I’ll talk about some of it below.

- The study has a small sample size. It is not clear if they chose this number because of expediency for because they achieved a saturation of themes.

- Table 1 should contain the participant characteristics. The current Table 1 (Acceptability constructs used) can instead be in the methods text or listed as an Annex. It would be better to include even more information, such as: years working, salaried or volunteers, years of education, years living in the community (birth or recent arrival), whether they have children and the average age of their children, etc. All of this paints a much richer picture of who these CHWs are. Not all CHWs are the same and we need to know more about these CHWs to understand their insights.

- We need to better understand if the CHWs are salaried or volunteers? The authors say that CHWs “are voluntary public health workers” in the introduction, but this isn’t always true; many countries across the globe and in Africa are forming national salaried cadres because evidence shows it is simply better (for outcomes, for economies, for equity, etc.).

- We need to better understand the specific tasks that they do? What is involved in the educational sessions and the support visits. There are different ways to implement such sessions/visits so the details matter.

- How many hours a week do they work?

- Main mechanism seems to be readily available information in communities (and even on weekends), with some cost savings because there were no user fees such as is charged in the clinic. Involving families likely also potentiates the intervention, and this has been described in the literature. In the setting of constrained budgets, nurses like the intervention because it allows for some shifting of tasks.

- The authors put a lot of weight on the Health Belief Model. This is a constrained understanding of the work that CHW programs can do, and I encourage the authors to reconsider a broader platform. The feedback from their own patient that education is insufficient speaks to this, and actually reflects best thinking about CHW interventions. It has been described that CHWs provide 4 main types of supports, among others: informational support (the education described here), emotional support (responding to affect, caring), instrumental support (helping access to services, medications, tools to support adherence such as pill boxes or reminder alarms, exercise classes, walking clubs, cooking classes), and material support (cash transfers, universal insurance without user fees, zero stockouts, medically tailored meals, gym membership). The authors don’t delineate these different inputs. They describe that in addition to access to information, there is an involvement of “social supports” (line 215) but what they consider social support doesn’t follow commonly agreed upon definitions, which more closely approximates the “material supports” mentioned above.

- As such, I urge the authors to recognize that the rise of NCDs is due to constrained choices in the face of poverty, not only due to poor lifestyle choices. These patients likely don’t always have access to healthy foods, or safe places to exercise. They eat sugary and salty foods because these are cheap ways to make food tasty. Trying to educate such a person about how to make better choices usually falls flat because they don’t have the tools to put that information into action.

- There is a lot that the authors can do to improve this intervention. I recommend that they look at the literature on:

o medically tailored meals (https://fimcoalition.org/research/existing-research/)

o adherence support (https://gh.bmj.com/content/3/1/e000566)

o improved access to healthy foods and exercise options (https://www.ncbi.nlm.nih.gov/pmc/articles/PMC9182982/)

- The PLA groups is also a fascinating example of how to structure community engagement in a more empowering way. I recommend that the authors recognize that this education campaign is a “top down” intervention – the assumption is that “the patients are without knowledge and the health system needs to fill that gap of knowledge via CHWs”. It is also the case that people often know a lot about their environment and only need support in organizing around feasible solutions. Such has been done in other contexts, and such groups are much better positioned to implement potentially major solutions, such as: vegetable gardening in community spaces, teaching classes on how to make food tasty without salt and sugar, applying community taxes on Sugary Beverages so as to encourage healthier options, clearing and guarding a space for patients to exercise safely, etc. These things can be very hard for an individual family to do, but may be possible with collective action (maybe even with a small cash support to advance preliminary plans). This example from Bangladesh is an example: https://www.ncbi.nlm.nih.gov/pmc/articles/PMC6830002/

- To improve the paper, I recommend the authors acknowledge these limitations and focus only on the single and minor intervention: education was generally acceptable and showed some benefits but is likely insufficient. I suspect their larger trial will show this too.

- Finally, the authors mention trust, but don’t

6. PLOS authors have the option to publish the peer review history of their article (what does this mean? ). If published, this will include your full peer review and any attached files.

**Do you want your identity to be public for this peer review?** For information about this choice, including consent withdrawal, please see our Privacy Policy .

Reviewer #1: **Yes: ** Tinashe Goronga

Reviewer #2: No

Reviewer #3: **Yes: ** Daniel Palazuelos, MD, MPH

---

## [Decision Letter · Decision Letter 1]

4 Dec 2024

PGPH-D-24-01557R1

Acceptability of a community health worker- led health literacy intervention on lifestyle modification among hypertensive and diabetes patients in the City of Harare, Zimbabwe

Dear Dr. Katena,

Thank you for submitting your manuscript to PLOS Global Public Health. After careful consideration, we feel that it has merit but does not fully meet PLOS Global Public Health’s publication criteria as it currently stands. Therefore, we invite you to submit a revised version of the manuscript that addresses the points raised during the review process.

You have done substantial and important work in revising the piece based on the initial reviewer comments, but one of the reviewers has some important follow up points that I agree should be addressed. Please address these and resubmit at your earliest convenience. 

We look forward to receiving your revised manuscript.

Kind regards,

Bram Wispelwey, MD, MS, MPH

Academic Editor

Journal Requirements:

Additional Editor Comments (if provided):

Reviewers' comments:

Reviewer's Responses to Questions

**Comments to the Author**

1. If the authors have adequately addressed your comments raised in a previous round of review and you feel that this manuscript is now acceptable for publication, you may indicate that here to bypass the “Comments to the Author” section, enter your conflict of interest statement in the “Confidential to Editor” section, and submit your "Accept" recommendation.

Reviewer #2: (No Response)

Reviewer #3: All comments have been addressed

2. Does this manuscript meet PLOS Global Public Health’s publication criteria ? Is the manuscript technically sound, and do the data support the conclusions? The manuscript must describe methodologically and ethically rigorous research with conclusions that are appropriately drawn based on the data presented.

Reviewer #2: Yes

Reviewer #3: Yes

3. Has the statistical analysis been performed appropriately and rigorously?

Reviewer #2: N/A

Reviewer #3: I don't know

4. Have the authors made all data underlying the findings in their manuscript fully available (please refer to the Data Availability Statement at the start of the manuscript PDF file)?

Reviewer #2: Yes

Reviewer #3: Yes

5. Is the manuscript presented in an intelligible fashion and written in standard English?

Reviewer #2: Yes

Reviewer #3: Yes

6. Review Comments to the Author

Reviewer #2: I wish to recognize the major amount of work that the authors have done in re-analyzing their qualitative data and rewriting/reformatting the bulk of the results section. The results are much better now, and I believe my remaining concerns (which are on things that have been changed since the first version) can easily be addressed through revision.

1) I do not think that the category of “Opportunity Costs” is being applied as defined by the Sekhon et al. conceptual framework. According to the framework, this concept is about tradeoffs that one must make to participate (that is, giving up something else in order to participate). The first subtheme in the table (“beliefs that participation in the intervention will consume time for other activities”) is a good example of this. However, the remaining subthemes in the table seem like they belong in “burden,” which Sekhon defines as encompassing costs of participation (from Sekhon reference: “e.g. participation requires too much time or expense, or too much cognitive effort”). Indeed, the second subtheme in opportunity costs (“additional resources will be required…”) seems the same as the first subtheme in burden (“concerns about need for additional resources…”). Alleviation from clinic-related costs described in the third and fourth subthemes of Opportunity Costs also seem to be about burden, not opportunity costs. And the results text focuses on perceived benefits and cost-effectiveness, which are not related to opportunity costs. I think the themes need to be reassigned and the results paragraph rewritten for this section on Opportunity Costs.

2) In Table 3, in the Perceived Effectiveness category, it is unclear how the empowerment subtheme is separate from the subtheme on increasing knowledge, skills and practice. The second and third quotes for empowerment are related to teaching/knowledge, and sound very similar to the quotes in the knowledge/skills/practices subtheme. There is also a quote in the middle of these two subthemes that is not clearly assigned to one or the other (Male patient IDI 10).

3) In Table 3, the second subtheme in Self-Efficacy would be better described in terms of feelings of confidence rather than ease of participation. Neither of the quotes suggests anything about the intervention being easy, and ease/difficulty of participation would reflect burden more than than self-efficacy in the framework. This distinction is particularly highlighted by Sekhon et al. (“The TFA construct of burden focuses on the burden associated with participating in the intervention (e.g. participation requires too much time or expense, or too much cognitive effort, indicating the burden is too great) rather than the individual’s confidence in engaging in the intervention (see definition of self-efficacy below).”) The results text aligns very well with self-efficacy – I think the description in the table just needs to be reworded.

Reviewer #3: The edits made are overall very good. While not all of my comments were fully taken, I think that this final product captures well what they are trying to do, and what they found in the study. I encourage the authors to continue thinking about partnering with CHWs as only part of the solution for combating NCDs. It will also be necessary to implement programs that give people material supports, such as better food, better medicines, etc. (as they recognized by including this in their discussion and conclusion).

7. PLOS authors have the option to publish the peer review history of their article (what does this mean? ). If published, this will include your full peer review and any attached files.

**Do you want your identity to be public for this peer review?** For information about this choice, including consent withdrawal, please see our Privacy Policy .

Reviewer #2: No

Reviewer #3: **Yes: ** Daniel Palazuelos, MD, MPH

---

## [Editor Report · Decision Letter 2]

19 Dec 2024

Acceptability of a community health worker- led health literacy intervention on lifestyle modification among hypertensive and diabetes patients in the City of Harare, Zimbabwe

PGPH-D-24-01557R2

Dear Dr. Katena,

We are pleased to inform you that your manuscript 'Acceptability of a community health worker- led health literacy intervention on lifestyle modification among hypertensive and diabetes patients in the City of Harare, Zimbabwe' has been provisionally accepted for publication in PLOS Global Public Health.

Best regards,

Bram Wispelwey, MD, MS, MPH

Academic Editor
